# Rhapontigenin Suppresses Leptin-Induced Vasculogenic Mimicry by Inhibiting STAT3-Aquaporin-1 Axis in TNBC Cells

**DOI:** 10.3390/biomedicines13123000

**Published:** 2025-12-07

**Authors:** Seung-Il Wang, Seung-Hyeon Lee, Eun-Ok Lee

**Affiliations:** 1Department of Science in Korean Medicine, College of Korean Medicine, Graduate School, Kyung Hee University, 26, Kyungheedae-ro, Dongdaemun-gu, Seoul 02447, Republic of Korea; seung8543@khu.ac.kr (S.-I.W.); leesh3788@khu.ac.kr (S.-H.L.); 2Department of Cancer Preventive Material Development, College of Korean Medicine, Graduate School, Kyung Hee University, 26, Kyungheedae-ro, Dongdaemun-gu, Seoul 02447, Republic of Korea

**Keywords:** rhapontigenin, leptin, vasculogenic mimicry, triple-negative breast cancer, aquaporin-1, signal transducer and activator of transcription 3

## Abstract

**Background/Objectives:** Vasculogenic mimicry (VM) is a process in which tumor cells form channel structures that resemble blood vessels in shape and function and is increasingly being recognized as a mechanism that contributes to triple-negative breast cancer (TNBC) progression and treatment resistance. Leptin, an adipokine that is elevated in patients with obesity, influences VM in breast cancer. Aquaporin-1 (AQP1), a water and solute channel, mediates leptin-induced VM. Rhapontigenin (Rha) is a stilbene derivative that exhibits diverse biological effects, including antioxidant, anti-inflammatory, and anticancer properties. This study investigated whether Rha inhibits leptin-induced VM and whether the mechanism involves AQP1 in TNBC cells. **Methods**: Cell viability was measured via MTT assay. mRNA and protein expression levels were measured by RT-qPCR and Western blot analysis, respectively. The DNA-binding activity of the signal transducer and activator of transcription 3 (STAT3) was determined using chromatin immunoprecipitation. Invasion and VM tube formation assays were performed. **Results**: Rha downregulated leptin-induced AQP1 mRNA and protein expression in TNBC cells without cytotoxicity. Phosphorylation of STAT3 by leptin was decreased after Rha treatment. Rha attenuated leptin-induced STAT3 DNA-binding activity at the AQP1 promoter. In addition, Rha inhibited leptin-induced invasion and VM. Consistent with these effects, the expression levels of invasion- and VM-related proteins and matrix metalloproteinase-2 activity were increased by leptin, which was reduced after Rha treatment. **Conclusions**: These results indicate that Rha inhibits invasive behavior and VM in TNBC cells by interfering with the leptin–STAT3–AQP1 signaling pathway, suggesting that Rha is a promising therapeutic candidate for the treatment of obesity-associated TNBC.

## 1. Introduction

Breast cancer is one of the most common malignancies in women and accounts for a substantial proportion of cancer-related mortality worldwide [1,2]. Breast cancer is clinically classified according to the expression of estrogen receptor, progesterone receptor, and human epidermal growth factor receptor 2. Tumors lacking all three markers are defined as triple-negative breast cancer (TNBC) [3]. Approximately 15–20% of breast cancer cases fall into the TNBC subtype, which is highly aggressive with few available therapeutic options. Patients with TNBC experience rapid progression, poor prognosis, and high metastatic potential [4,5]. Despite advances in screening and therapy, its incidence continues to increase [6]. Therefore, there is an urgent need to better understand the molecular mechanisms underlying TNBC progression and to identify novel therapeutic strategies.

Obesity is a risk factor for breast cancer and contributes to tumor growth and shorter disease-free survival [7,8,9]. Leptin, an adipokine elevated in patients with obesity, plays an important role in energy homeostasis [7,10]. Beyond its classical role, leptin promotes cancer progression including angiogenesis and metastasis; by activating oncogenic pathways, such as the JAK2/STAT3 axis; in breast cancer [7,11]. Leptin has also been identified as a regulator of cellular invasiveness and VM in TNBC cells [11,12,13]. This suggests that obesity-associated hyperleptinemia may accelerate TNBC progression and may be associated with a poor prognosis.

Vasculogenic mimicry (VM) is the ability of tumor cells to form extracellular matrix–rich vessel-like channels that conduct blood flow independently of endothelial cells. These structures supply nutrients and oxygen, which are essential for tumor growth, and promote metastatic spread [14,15]. Since its first report in melanoma, VM has been observed in several solid tumors, including breast cancer, where it is correlated with poor prognosis, therapeutic resistance, and enhanced metastatic potential [16]. Notably, in TNBC mouse models, discontinuation of anti-angiogenic therapy paradoxically promotes VM formation and tumor invasiveness, suggesting that tumor hypoxia induced by anti-angiogenic pressure drives VM as an adaptive survival mechanism [17]. Collectively, VM represents an alternative vascularization strategy and a compelling therapeutic target beyond conventional antiangiogenic approaches.

Aquaporin-1 (AQP1), a water and solute channel, has been implicated not only in fluid homeostasis but also in cancer progression [18,19]. Animal studies have demonstrated that AQP1 facilitates angiogenesis and that its inhibition significantly suppresses melanoma tumor growth in mice [20,21]. AQP1 interacts with key intracellular signaling pathways, suppressing apoptosis while amplifying pro-migratory and pro-invasive signals, thereby promoting tumor progression [22]. In addition to conventional angiogenesis, AQP1 has been associated with the VM process [22,23]. Overexpression of AQP1 promotes VM and metastasis in breast cancer, whereas its suppression inhibits VM, highlighting its critical role in tumor aggressiveness [12,24,25]. Thus, targeting AQP1 may provide therapeutic benefits by disrupting both VM and metastatic capacity [26,27].

Natural compounds with multitarget activities have recently attracted attention as potential anticancer agents [28]. Rhapontigenin (Rha), a stilbene derivative, exerts diverse biological effects including antioxidant, anti-inflammatory, and anticancer properties [29,30,31]. Rha suppresses angiogenesis by inhibiting epithelial–mesenchymal transition and modulating hypoxia-inducible factor-1 signaling [32,33]. However, its role in VM regulation remains largely unknown. To date, there have been no reports on the relationship between Rha and VM. In our previous study, we demonstrated that leptin induces VM via the STAT3-AQP1 pathway in TNBC cells [12]. Therefore, we hypothesized that Rha might suppress VM by modulating this pathway. The present study aims to evaluate the inhibitory effects of Rha on leptin-induced VM in TNBC cells and to explore its potential as a novel therapeutic candidate for obesity-associated TNBC.

## 2. Materials and Methods

### 2.1. Cell Culture

MDA-MB-231 and Hs 578T human TNBC cell lines were procured from the Korean Cell Line Bank (Seoul, Republic of Korea). MDA-MB-231 cells were grown in RPMI 1640 (WELGENE, Daegu, Republic of Korea), and Hs 578T cells were grown in DMEM (WELGENE) supplemented with 10% fetal bovine serum (WELGENE) and 1% antibiotics (WELGENE). All cultures were maintained in a 37 °C incubator with 5% CO_2_ and saturated humidity.

### 2.2. Cell Viability Assay

Cell viability was assessed using a 3-(4,5-dimethylthiazol-2-yl)-2,5-diphenyltetrazolium bromide (MTT; Sigma-Aldrich, St Louis, MO, USA) assay. MDA-MB-231 and Hs 578T cells were seeded at a density of 2 × 10^4^ cells per well in a 96-well plate and incubated for 24 h. Cells were treated with Rha (Tokyo Chemical Industry, Tokyo, Japan, purity: ≥ 98% by GC) at the indicated concentrations. After incubation for 24 h, 50 μL of MTT solution (1 mg/mL) was added to each well and incubated for 2 h. Formazan crystals were dissolved with 100 μL of DMSO (Sigma-Aldrich), and absorbance was measured at 570 nm (Sunrise RC, TECAN, Salzburg, Austria).

### 2.3. RNA Isolation and Real-Time Polymerase Chain Reaction (RT-qPCR)

Total RNA was extracted using TRIzol reagent (Invitrogen, Carlsbad, CA, USA). RNA (2 μg) was reverse transcribed into cDNA, and qPCR was performed using GreenStar™ qPCR Master Mix (Bioneer Corporation, Daejeon, Republic of Korea) and specific primers on a Thermal Cycler Dice^®^ Real Time System III (Takara, Katsushika City, Tokyo, Japan). The primer sequences were as follows: AQP1: forward 5′-CAGCCCAAGGACAGTTCAGAG-3′; reverse 5′-CCATCATGGCTAAGTGCACAG-3′ β-actin: forward 5′-AAGAGAGGCATCCTCACCCT-3′; reverse 5′-ATCTCTTGCTCGAAGTCCAG-3’.

### 2.4. Western Blot Analysis

Cell lysates (30 μg protein) were separated by 8–12% SDS-PAGE, transferred to nitrocellulose membranes (Pall Corporation, Port Washington, NY, USA), and blocked with 5% skim milk. The membranes were incubated overnight at 4 °C with primary antibodies listed in Table 1, followed by secondary antibody incubation for 2 h at room temperature. Protein bands were visualized using an enhanced chemiluminescence reagent (GE Healthcare, Chicago, IL, USA).

### 2.5. Chromatin Immunoprecipitation (ChIP) Assay

ChIP assays were performed using the SimpleChIP Enzymatic Chromatin IP Kit (Cell Signaling Technology) according to the manufacturer’s instructions. Cells were treated with leptin (R&D Systems, Minneapolis, MN, USA, 100 ng/mL) alone or with Rha for 24 h, cross-linked with 1% formaldehyde, and sonicated. Immunoprecipitation was performed using a STAT3 antibody or negative control IgG. Purified DNA was analyzed by qPCR using AQP1 promoter-specific primers: forward 5′-CCTGAGTTCAGTGGCTCCTTG-3′; reverse 5′-CACCTGTCCCTGCCCTGCTA-3′.

### 2.6. Matrix Metalloproteinase-2 (MMP-2) Activity Assay

MMP-2 activity was measured using the Human MMP-2 Activity Assay Kit (QuickZyme Biosciences, Leiden, The Netherlands). Cells were seeded into a 24-well plate and treated with leptin (100 ng/mL) alone or with Rha for 24 h. The conditioned culture medium was collected, and MMP-2 activity was measured according to the manufacturer’s protocol.

### 2.7. Invasion Assay

The invasion assay was performed using a Transwell system (Costar^®^ Transwell^®^ cell culture inserts, 8 μM pore size; Corning Inc., Corning, NY, USA) coated with 20-fold diluted Matrigel (BD Biosciences, San Jose, CA, USA). MDA-MB-231 cells (2.5 × 10^5^) or Hs 578T cells (4.0 × 10^5^) with Rha (MDA-MB-231: 30 µM, Hs 578T: 20 µM) were seeded into the upper chamber and leptin (100 ng/mL) was added to the lower chamber. After 24 h of incubation, cells were fixed and stained. Non-invading cells in the upper chamber were removed and invading cells were imaged under an optical microscope (Ts2_PH, Nikon, Tokyo, Japan) at 200× magnification.

### 2.8. Three-Dimensional (3D) Culture VM Tube Formation Assay

VM was assessed in Matrigel-coated 24-well plates. After Matrigel (BD Biosciences) polymerization at 37 °C for 1 h, MDA-MB-231 cells (1.5 × 10^5^) or Hs 578T cells (4.0 × 10^5^) were seeded under three conditions: control, leptin (100 ng/mL), and leptin combined with Rha (MDA-MB-231: 30 μM; Hs 578T: 20 μM). After 16 h of incubation, the tube structures were imaged using an optical microscope (Nikon, Tokyo, Japan) at 40× magnification.

### 2.9. Statistical Analysis

All data were reported as mean ± standard deviation (SD). Experiments were performed in triplicate. Statistical analysis employed ANOVA followed by Tukey’s post hoc tests for multiple group comparisons using GraphPad Prism software (version 5, GraphPad Software Inc., Boston, MA, USA). Different letters on the graph indicate statistically significant differences between groups.

## 3. Results

### 3.1. Rhapontigenin Downregulated Leptin-Induced AQP1 Expression at a Non-Cytotoxic Concentration in TNBC Cells

To determine the appropriate concentrations for subsequent experiments, the viability of Rha-treated MDA-MB-231 and Hs578T cells was assessed using the MTT assay. Rha did not significantly affect cell viability at concentrations up to 30 µM in MDA-MB-231 cells and up to 20 µM in Hs 578T cells, but higher concentrations led to a marked reduction in viability (Figure 1A,B). Based on these findings, 30 µM for MDA-MB-231 cells and 20 µM for Hs578T cells were selected as non-cytotoxic concentrations for further experiments to ensure that observed molecular or functional changes were not due to general cytotoxicity.

To evaluate the effect of Rha on the expression of AQP1, cells were treated with leptin alone or in combination with Rha for 24 h. AQP1 mRNA expression was quantified by RT-qPCR, and protein expression was analyzed by Western blot. Leptin treatment alone led to a significant upregulation of AQP1 mRNA in both cell lines compared to that in the untreated controls. Co-treatment with Rha significantly attenuated the leptin-induced increase in AQP1 expression (Figure 1C,D, Table A1 and Table A2). Western blotting confirmed these findings at the protein level. Consistent with the RT-qPCR results, leptin enhanced AQP1 protein expression, whereas co-treatment with Rha markedly suppressed this effect (Figure 1E,F). Collectively, these results demonstrated that Rha downregulates leptin-induced AQP1 expression in TNBC cells at non-cytotoxic concentrations.

### 3.2. Rhapontigenin Impaired Leptin-STAT3 Signaling in TNBC Cells

To investigate whether the downregulation of AQP1 protein expression by Rha was mediated by STAT3, Western blot analysis, ChIP assay, and qPCR were performed. First, cells were treated with leptin alone or in combination with Rha for 30 min, and STAT3 phosphorylation was assessed by Western blot. Leptin markedly increased the levels of phosphorylated STAT3, whereas co-treatment with Rha significantly suppressed this phosphorylation without altering total STAT3 protein levels in both TNBC cell lines (Figure 2A,B). Second, cells were treated with leptin alone or in combination with Rha for 24 h. Leptin alone significantly enhanced the DNA-binding activity of STAT3 at the AQP1 promoter, showing a 3-fold increase compared to untreated controls in MDA-MB-231 cells. However, this effect was significantly inhibited by the presence of Rha (Figure 2C, Table A3). Consistent with the results in MDA-MB-231 cells, the same effect was observed in Hs 578T cells. A 6-fold increase in the DNA-binding activity of STAT3 by leptin, compared with untreated controls, was reduced in the presence of Rha (Figure 2D, Table A4). Taken together, these results suggested that Rha downregulates the expression of AQP1 by inhibiting STAT3 activity and the DNA-binding activity of STAT3 at the AQP1 promoter in TNBC cells.

### 3.3. Rhapontigenin Suppressed Leptin-Induced Invasion and VM in TNBC Cells

To examine the functional effect of Rha on cellular invasiveness, a Matrigel-coated Transwell invasion assay was conducted. Cells containing Rha were seeded in the upper chamber, and medium containing leptin was added to the lower chamber, followed by incubation for 24 h. Leptin treatment alone significantly increased the invasive ability of MDA-MB-231 cells, showing a 3-fold increase compared to untreated controls. In contrast, co-treatment with Rha substantially reduced the number of invading cells induced by leptin (Figure 3A). Consistent with the results in MDA-MB-231 cells, leptin-induced invasion of Hs 578T cells was significantly reduced in the presence of Rha (Figure 3B). These findings demonstrated that Rha effectively inhibits the invasive ability of both leptin-treated TNBC cell lines.

Next, to evaluate whether Rha inhibits leptin-induced VM, TNBC cells were seeded into Matrigel-coated wells and treated with leptin alone or in combination with Rha for 16 h to allow the formation of a three-dimensional vascular-like network. Following leptin treatment, the MDA-MB-231 cells exhibited a significant increase in the number of tube-like structures with complex branching. Co-treatment with Rha markedly reduced both the number and complexity of these structures, demonstrating a potent inhibitory effect on VM (Figure 4A). Consistent with the results in MDA-MB-231 cells, the inhibitory effect of Rha on the VM network structure induced by leptin was observed in Hs 578T cells (Figure 4B). These findings suggested that Rha interferes with the cellular processes underlying VM and effectively counteracts leptin-mediated VM in TNBC cells.

Taken together, these results demonstrated that Rha suppresses leptin-induced aggressive tumor phenotypes, such as invasiveness and VM, in TNBC cells.

### 3.4. Rhapontigenin Inhibited Leptin-Induced Invasion- and VM-Related Proteins in TNBC Cells

To demonstrate the role of Rha in invasion- and VM-related proteins, TNBC cells were treated with leptin alone or in combination with Rha for 24 h and Western blot analysis was performed. LAMC2, VE-cadherin, and MMP-2 protein expression was upregulated by leptin treatment alone. In contrast, Rha co-treatment markedly reduced the expression of these proteins in both TNBC cell lines (Figure 5A,B). Additionally, MMP-2 enzymatic activity was measured in the conditioned medium from cells treated with leptin alone or in combination with Rha for 24 h using a commercial activity assay kit. MMP-2 activity was significantly elevated following leptin treatment, whereas Rha effectively suppressed this increase in both TNBC cell lines (Figure 5C,D). These results indicated that Rha inhibits leptin-induced invasion and VM in both TNBC cell lines by reducing the expression and activity of invasion- and VM-related proteins.

## 4. Discussion

VM is increasingly recognized as a key mechanism by which aggressive tumor cells sustain their blood supply and metastatic potential, independent of endothelial cell-mediated angiogenesis [14,15]. Beyond serving as an alternative vascularization strategy, VM has been identified as a major contributor to therapeutic resistance to conventional anti-angiogenic agents, such as VEGF-targeting therapies [34,35]. VM drives TNBC progression through various signaling pathways and represents a barrier to effective targeted therapy [36]. Therefore, it is crucial to identify the signaling pathways that regulate VM and develop strategies to inhibit them.

Leptin is an adipokine secreted by adipose tissue, and its primary function is to regulate energy balance and body weight [37]. Leptin has been implicated in the link between obesity and breast cancer [38] and is associated with tumor progression, invasion, and treatment resistance in breast cancer [39]. A recent study showed that leptin induces VM through the leptin receptor/STAT3 pathway in TNBC cells such as MDA-MB-231 and Hs 578T cells, and AQP1 mediates this effect [12]. Therefore, in this study, we focused on the leptin–STAT3–AQP1 axis to assess whether Rha could effectively suppress leptin-induced VM in TNBC cells.

To demonstrate that the effects of Rha observed in this study were not due to cytotoxicity, a non-toxic concentration was established through cell viability experiments in MDA-MB-231 cells (Figure 1A) and Hs 578T cells (Figure 1B), and subsequent experiments were conducted accordingly. RT-qPCR and Western blot analyses revealed that Rha downregulated leptin-induced AQP1 mRNA (Figure 1C,D) and protein (Figure 1E,F) levels at nontoxic concentrations. Therefore, this inhibition appears to be due to the disruption of a specific pathway rather than cytotoxic stress, highlighting AQP1 as a functional downstream target of Rha and supporting the hypothesis that Rha effectively disrupts this signaling axis. To elucidate the underlying mechanism, Western blot analyses were performed to determine the effect of Rha on the STAT3 signaling pathway. The overexpression and constitutive activation of STAT3 play critical roles in cell survival, progression, metastasis, and chemoresistance in TNBC. Several cytokines and growth factors activate STAT3 via phosphorylation [40]. Leptin-activated STAT3 signaling is well known to promote the growth and progression of breast cancer [41]. Consistent with recent results [13], Western blot revealed that leptin triggered STAT3 phosphorylation in TNBC cells. However, Rha effectively suppressed this phosphorylation without altering the total level of STAT3 (Figure 2A,B), suggesting that Rha selectively inhibits STAT3 activation rather than reducing STAT3 expression. Phosphorylated STAT3 forms STAT3 dimers, which then translocate from the cytoplasm to the nucleus, where they bind to specific DNA sequences in target gene promoters and act as an oncogenic transcription factor [40,42]. After confirming that Rha downregulated AQP1 expression at the transcriptional level (Figure 1C,D), a ChIP assay was performed to determine whether STAT3 bound to the AQP1 promoter. Figure 2C,D showed that leptin significantly enhanced the DNA-binding activity of STAT3 to the AQP1 promoter, which was suppressed by Rha treatment. This indicates that the leptin-STAT3 axis regulates AQP1 expression and that Rha interferes with this transcriptional regulation. These findings are consistent with previous studies that identified phosphorylated STAT3 as a central regulator of tumor-promoting gene expression in TNBC [43].

The leptin–STAT3–AQP1 signaling pathway contributes to the induction of VM in TNBC cells [12]. AQP1 promotes the migration and invasion of breast cancer cells [44]. Therefore, to verify the functional effects of Rha on the invasive ability and VM formation in TNBC, invasion and VM tube formation assays were conducted. As expected, Rha markedly reduced leptin-induced invasive behavior in both TNBC cell lines (Figure 3). Consistent with these results, Rha effectively impaired the formation of the vessel-like structures (Figure 4). Taken together, these results highlight the potential of Rha as a therapeutic agent capable of targeting aggressive tumor phenotypes such as invasiveness and VM. Leptin treatment increased the expression of invasion- and VM-related proteins in TNBC cells, which was reduced by Rha treatment (Figure 5A,B). Furthermore, Rha inhibited the elevated MMP-2 enzymatic activity induced by leptin (Figure 5C,D). VE-cadherin is essential for VM channel stability [45], LAMC2 mediates extracellular matrix (ECM)–tumor interactions, and MMP-2 facilitates ECM degradation to enable invasion and channel formation [46]. These molecular changes provide a mechanistic basis for the observed reduction in cell invasion and VM formation. These findings reveal a consistent sequence: leptin activates STAT3, inducing the expression of AQP1 and invasion- and VM-related proteins, which promote ECM remodeling, invasive behavior, and VM channel formation. Rha effectively suppressed VM in TNBC cells by interfering with this cascade at multiple levels, from transcriptional regulation to downstream stromal remodeling.

Obesity-associated TNBC is characterized by elevated circulating leptin levels, which have been linked to metabolic status and tumor aggressiveness [47]. Therefore, Rha may be particularly beneficial in patients with VM-positive obesity-associated breast cancer. Furthermore, combination therapy with Rha and conventional anti-angiogenic agents may address VM as a critical mechanism for therapeutic resistance.

## 5. Conclusions

This study provided the first evidence that Rha suppresses VM in TNBC cells by inhibiting leptin-induced STAT3 activation and AQP1 expression (Figure 6). This disruption of VM has been attributed to a reduction in invasion- and VM-associated proteins, ECM remodeling, and invasive behavior. These results highlight the therapeutic relevance of targeting the leptin–STAT3–AQP1 axis and suggest that Rha is a promising candidate for treating patients with TNBC, particularly in the context of obesity. However, further studies are needed, including an evaluation the VM-inhibiting activity of Rha using animal models and an evaluation of the efficacy of combination therapy with anti-angiogenic agents. By interfering with multiple essential steps in VM formation, Rha offers a novel strategy for overcoming leptin-induced aggressiveness and resistance to conventional anti-angiogenic therapies.

## Figures and Tables

**Figure 1 biomedicines-13-03000-f001:**
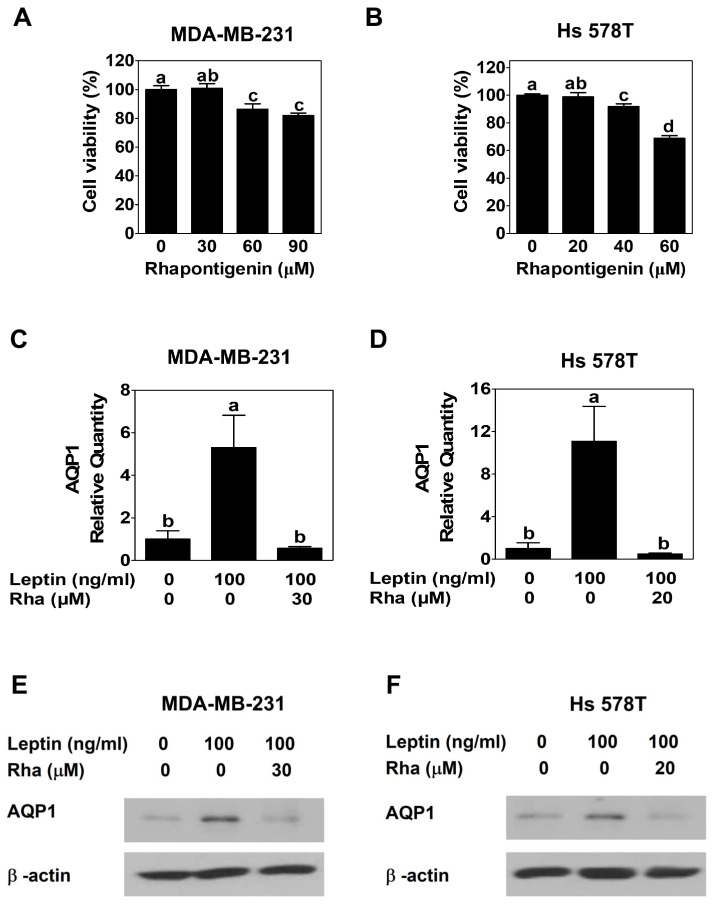
Rhapontigenin downregulated leptin-induced AQP1 expression at non-cytotoxic concentrations in TNBC cells. (**A**) MDA-MB-231 cells or (**B**) Hs 578T cells were exposed to Rha for 24 h. MTT assay was performed to determine cell viability in response to Rha. (**C**,**E**) MDA-MB-231 cells or (**D**,**F**) Hs 578T cells were treated with leptin alone or in combination with Rha for 24 h. (**C**,**D**) RT-qPCR was performed using RNA extracted from these cells. (**E**,**F**) Cell lysates were analyzed by Western blot. All data are expressed as mean ± SD from three independent experiments, Statistical analysis employed ANOVA followed by Tukey’s post hoc tests for multiple group comparisons. Different letters on the graph indicate statistically significant differences between groups.

**Figure 2 biomedicines-13-03000-f002:**
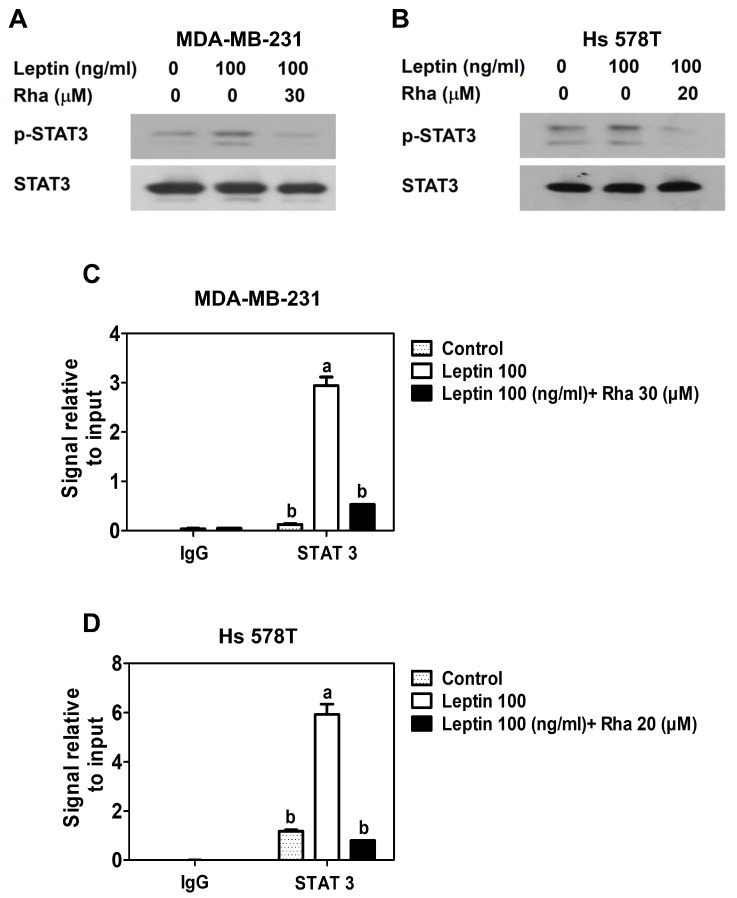
Rhapontigenin impaired leptin-STAT3 signaling in TNBC cells. (**A**) MDA-MB-231 or (**B**) Hs 578T cells were treated with leptin alone or in combination with Rha for 30 min and cell lysates were analyzed by Western blot. (**C**) MDA-MB-231 or (**D**) Hs 578T cells were treated with leptin alone or in combination with Rha for 24 h. ChIP assay was performed using STAT3 antibody or negative control IgG followed by qPCR. All data were expressed as mean ± SD from three independent experiments, Statistical analysis employed ANOVA followed by Tukey’s post hoc tests for multiple group comparisons. Different letters on the graph indicate statistically significant differences between groups.

**Figure 3 biomedicines-13-03000-f003:**
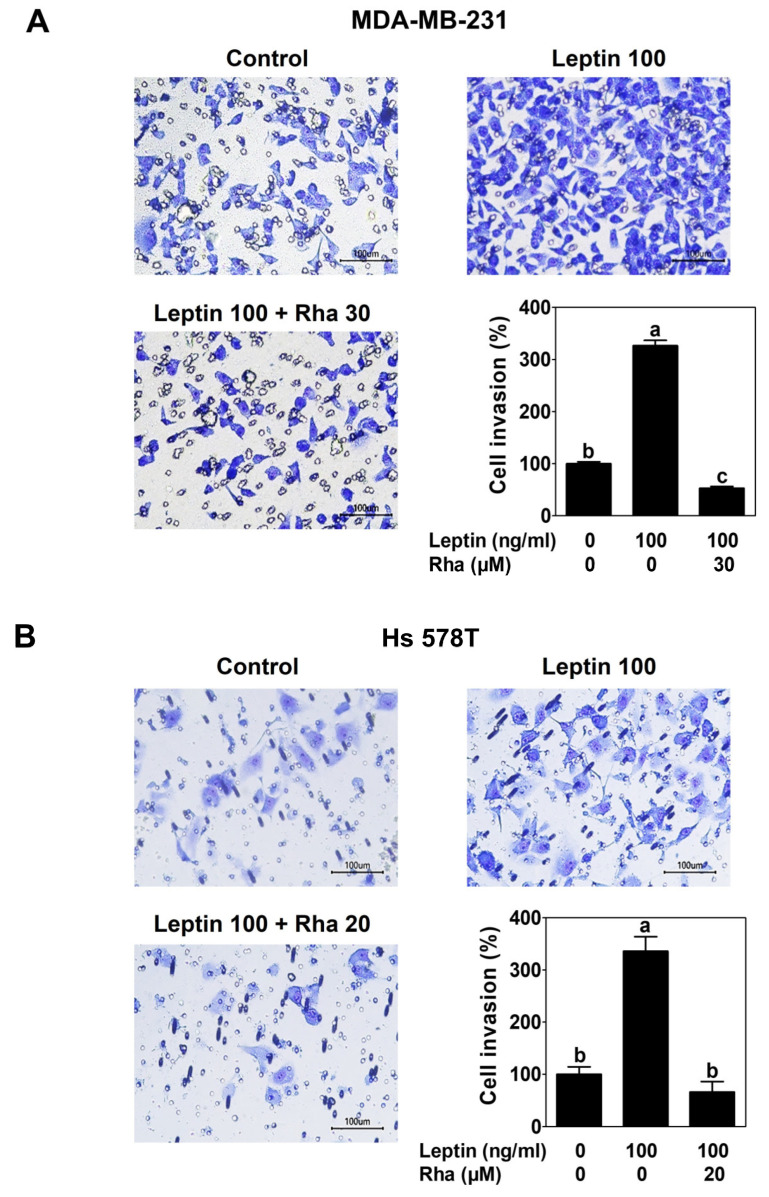
Rhapontigenin suppressed leptin-induced invasion in TNBC cells. Invasion assay was performed using a Matrigel-coated Transwell system. (**A**) MDA-MB-231 or (**B**) Hs 578T cells with Rha were seeded into the upper chamber. Leptin was used as a chemoattractant. After 24 h incubation, the invading cells were imaged under an optical microscope at 200× magnification. Scale bar = 100 μm. The number of invading cells was counted. All data were expressed as mean ± SD from three independent experiments. Statistical analysis employed ANOVA followed by Tukey’s post hoc tests for multiple group comparisons. Different letters on the graph indicate statistically significant differences between groups.

**Figure 4 biomedicines-13-03000-f004:**
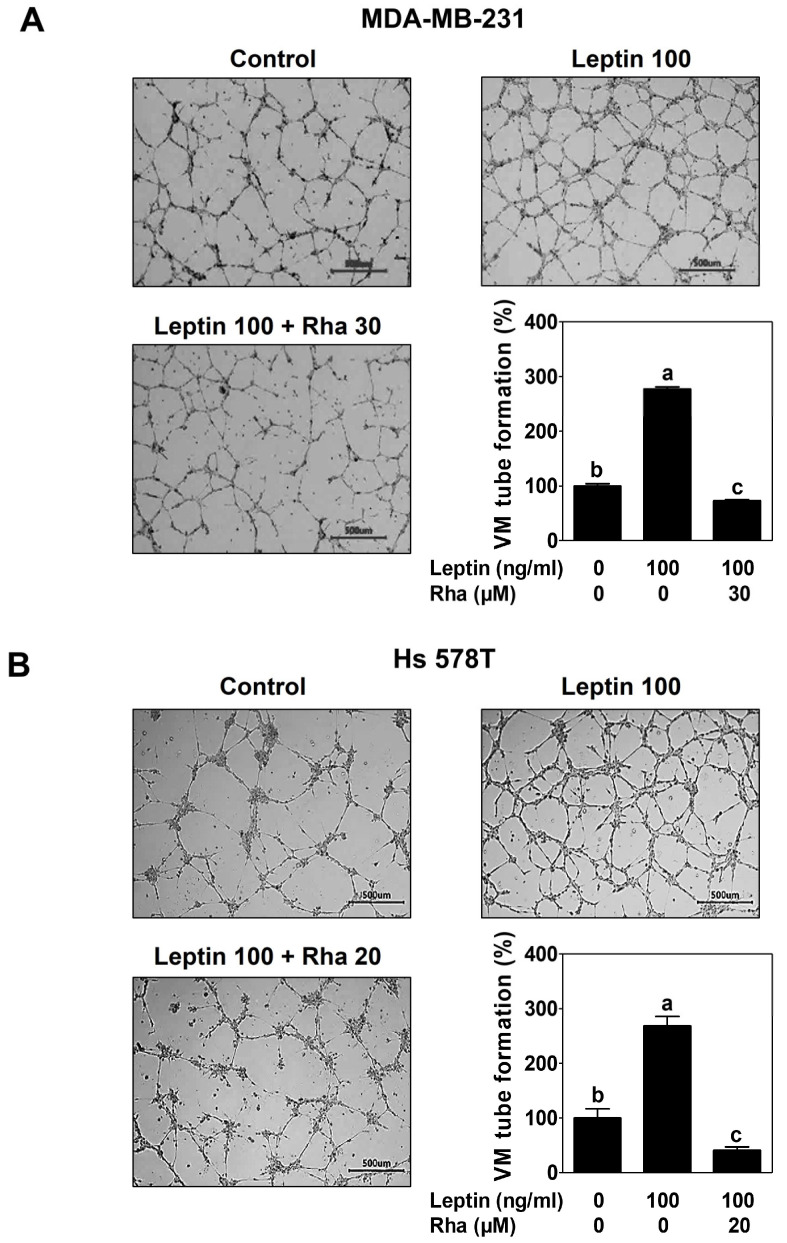
Rhapontigenin suppressed leptin-induced VM in TNBC cells. VM tube formation assay was conducted in Matrigel-coated wells. (**A**) MDA-MB-231 or (**B**) Hs 578T cells were treated with leptin alone or in combination with Rha for 16 h. VM structure was imaged under an optical microscope at 40× magnification. Scale bar = 500 μm. VM formation was calculated by counting the number of single circular structures formed by connecting three or more (cells) points. All data were expressed as mean ± SD from three independent experiments. Statistical analysis employed ANOVA followed by Tukey’s post hoc tests for multiple group comparisons. Different letters on the graph indicate statistically significant differences between groups.

**Figure 5 biomedicines-13-03000-f005:**
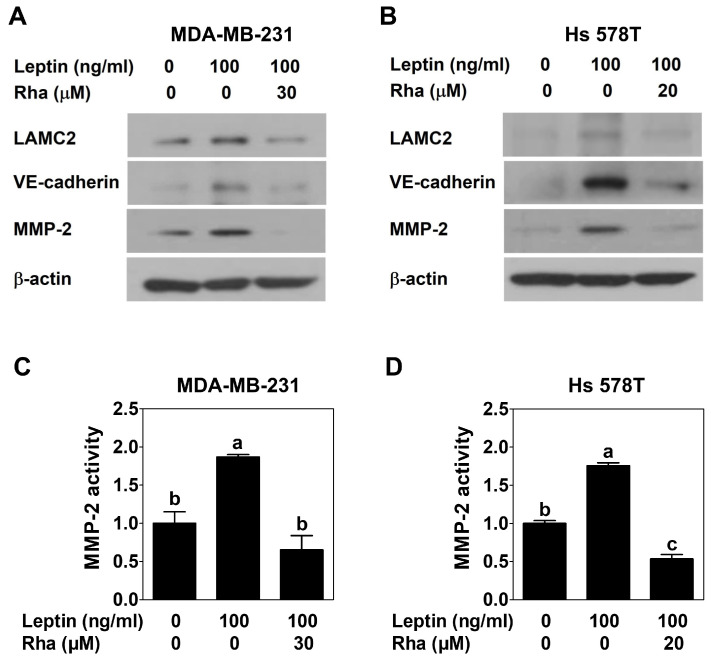
Rhapontigenin inhibited leptin-induced invasion- and VM-related proteins in TNBC cells. (**A**,**C**) MDA-MB-231 cells or (**B**,**D**) Hs 578T cells were treated with leptin alone or in combination with Rha for 24 h. (**A**,**B**) Cell lysates were analyzed by Western blot. (**C**,**D**) Conditioned culture medium was collected, and MMP-2 activity was measured. All data are expressed as mean ± SD from three independent experiments, Statistical analysis employed ANOVA followed by Tukey’s post hoc tests for multiple group comparisons. Different letters on the graph indicate statistically significant differences between groups.

**Figure 6 biomedicines-13-03000-f006:**
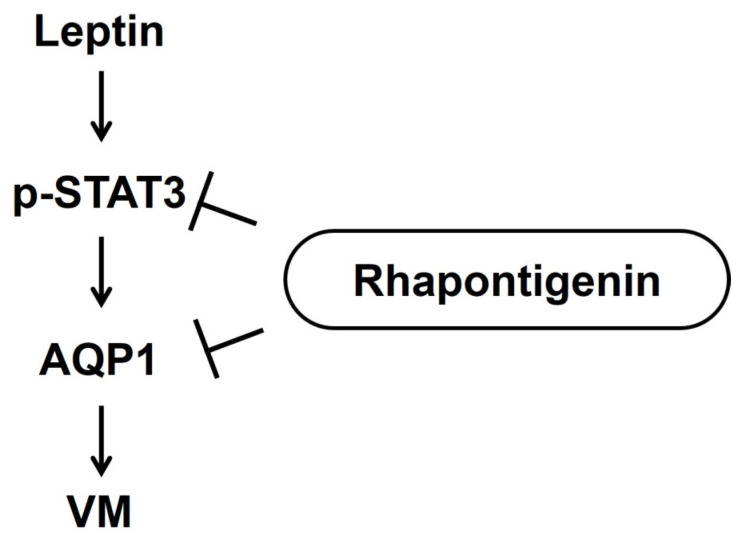
Schematic diagram of the inhibitory effects of rhapontigenin on leptin-induced vasculogenic mimicry in TNBC cells.

**Table 1 biomedicines-13-03000-t001:** Antibodies used in this study.

Antibody	Company	Dilution	Product No.
AQP1	Santa Cruz	1:500	SC-25287
β-actin	Sigma-Aldrich	1:20,000	A5316
pSTAT3	Cell Signaling	1:1000	9145S
STAT3	Cell Signaling	1:10,000	12640S
MMP-2	Abcam	1:1000	ab86607
LAMC2	Abcam	1:500	ab96327
VE-cadherin	Abgent	1:500	AP2724a
goat anti-rabbit IgG-HRP	Millipore	1:5000	AP187P
goat anti-mouse IgG-HRP	Millipore	1:5000	AP181P

Santa Cruz Biotechnology (Danvers, MA, USA); Sigma-Aldrich (St. Louis, MO, USA); Cell Signaling Technology (Beverly, MA, USA); Abcam (Cambridge, UK); Abgent (San Diego, CA, USA); and Millipore (Billerica, MA, USA).

## Data Availability

The data is contained within the article.

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
