# Peer review of "Rhapontigenin Suppresses Leptin-Induced Vasculogenic Mimicry by Inhibiting STAT3-Aquaporin-1 Axis in TNBC Cells"

_biomedicines, 2025, doi:10.3390/biomedicines13123000_

Round 1
Reviewer 1 Report
Comments and Suggestions for Authors
This study reports an inhibitory effect of Rha on the Leptin-STAT3–AQP1-VM pathway in TNBC. In my assessment, the manuscript currently suffers from a limited dataset, which inadequately supports the observed phenomena and the proposed mechanism. Furthermore, the logical flow of the narrative requires more thoughtful organization. The following specific points are raised for the authors' consideration:
- The term "regulating" in the title is ambiguous. The title should clearly reflect the study's conclusion, specifying whether the effect is activation or inhibition.
- The introduction should provide a more detailed background. For instance:
- Why was the relationship between Rha and VM chosen for investigation?
- Have there been any prior studies in other disease models suggesting a link between Rha and VM?
- Does this study represent the first report of this relationship?
- The introduction should explicitly cite literature that establishes the "Leptin-STAT3–AQP1-VM" axis as a recognized model for VM induction. This foundational citation is necessary to justify the experimental design in Figures 1-2 (e.g., "Leptin induces increased AQP1 transcription/STAT3 levels"). Simply introducing separate connections (Leptin-VM, AQP1-VM) is insufficient.
- A critical gap is the lack of experimental validation demonstrating that AQP1 or STAT3 are functionally involved in "leptin-induced VM" in the authors' model. To establish the proposed Leptin–STAT3–AQP1–VM pathway, it is essential to show that knockdown/inhibition of STAT3 activity or AQP1 expression blocks leptin-induced VM.
- Suggested experiment: Conditionally knock down/inhibit AQP1 expression or STAT3 activity and demonstrate that this reduces VM in the "leptin-VM" model. Only after establishing the crucial role of STAT3-AQP1 in VM, would investigating Rha's effects on them become meaningful.
- In Fig 5, Rha reduces both leptin-induced MMP-2 protein levels and its activity in the conditioned medium. The observed decrease in activity is likely a direct consequence of the reduced protein secretion.
- This hypothesis could be confirmed by quantifying secreted MMP-2 protein levels using ELISA.
- Therefore, the conclusion stating that Rha reduces "expression and activity" is not fully supported by the data. To claim a specific effect on "activity," enzymatic activity should be assessed relative to the protein amount (e.g., specific activity).
- The manuscript identifies multiple potential targets of Rha (AQP1 expression, STAT3 phosphorylation, MMP-2 levels/activity). For each of these, definitive rescue experiments are lacking. It is crucial to demonstrate whether the inhibitory effect of Rha on VM can be reversed (rescued) by constitutively activating the respective downstream components.
- The logical sequence of the figures could be improved. The phenotypic experiments validating the "leptin-VM" model (Figs 3, 4, 5) would be more logically placed at the beginning of the results. The mechanistic investigations in Figs 1 and 2, which should build upon the established phenotype, require more robust data for support.
Author Response
Comments 1 : The term "regulating" in the title is ambiguous. The title should clearly reflect the study's conclusion, specifying whether the effect is activation or inhibition.
Response 1 : Thank you for your valuable comments. According to your comments, we have changed the title as follows: Rhapontigenin Suppresses Leptin-induced Vasculogenic Mimicry by Inhibiting STAT3-Aquaporin-1 axis in TNBC Cells.
Comments 2 : The introduction should provide a more detailed background. For instance:
- Why was the relationship between Rha and VM chosen for investigation?
- Have there been any prior studies in other disease models suggesting a link between Rha and VM?
- Does this study represent the first report of this relationship?
Response 2 : We have revised the introduction to reflect your comments. This study represented the first report of this relationship. To date, there have been no reports on the relationship between rha and VM.
Comments 3 : The introduction should explicitly cite literature that establishes the "Leptin-STAT3–AQP1-VM" axis as a recognized model for VM induction. This foundational citation is necessary to justify the experimental design in Figures 1-2 (e.g., "Leptin induces increased AQP1 transcription/STAT3 levels"). Simply introducing separate connections (Leptin-VM, AQP1-VM) is insufficient.
Response 3 : According to your comments, we explicitly cited the literature establishing the “leptin-STAT3–AQP1-VM” axis as a recognized model for VM induction in the introduction. [Han, D. S.; Lee, E. O., Leptin Promotes Vasculogenic Mimicry in Breast Cancer Cells by Regulating Aquaporin-1. Int. J. Mol. Sci. 2024, 25, (10).]
Comments 4 : A critical gap is the lack of experimental validation demonstrating that AQP1 or STAT3 are functionally involved in "leptin-induced VM" in the authors' model. To establish the proposed Leptin–STAT3–AQP1–VM pathway, it is essential to show that knockdown/inhibition of STAT3 activity or AQP1 expression blocks leptin-induced VM.
- Suggested experiment: Conditionally knock down/inhibit AQP1 expression or STAT3 activity and demonstrate that this reduces VM in the "leptin-VM" model. Only after establishing the crucial role of STAT3-AQP1 in VM, would investigating Rha's effects on them become meaningful.
Response 4 : In our previous paper [Han, D. S.; Lee, E. O., Leptin Promotes Vasculogenic Mimicry in Breast Cancer Cells by Regulating Aquaporin-1. Int. J. Mol. Sci. 2024, 25, (10).], we have already confirmed the leptin-STAT3-AQP1-VM axis using STAT3 inhibitors and AQP1 overexpression/siRNA, and described this in the Introduction and Discussion sections.
Comments 5 : In Fig 5, Rha reduces both leptin-induced MMP-2 protein levels and its activity in the conditioned medium. The observed decrease in activity is likely a direct consequence of the reduced protein secretion.
- This hypothesis could be confirmed by quantifying secreted MMP-2 protein levels using ELISA.
- Therefore, the conclusion stating that Rha reduces "expression and activity" is not fully supported by the data. To claim a specific effect on "activity," enzymatic activity should be assessed relative to the protein amount (e.g., specific activity).
Response 5 : That's correct. However, MMP-2 protein amount and activity do not always directly correlate, and because MMP-2 activity is more important than protein amount, we used a kit to assess activity. Since the results of activity assay kit are expressed as mmp-2 activity in other papers, we also expressed as activity in this paper. It would have been better to quantify MMP-2 secretion using ELISA, but as an alternative, we presented protein amounts using Western blot.
Comments 6 : The manuscript identifies multiple potential targets of Rha (AQP1 expression, STAT3 phosphorylation, MMP-2 levels/activity). For each of these, definitive rescue experiments are lacking. It is crucial to demonstrate whether the inhibitory effect of Rha on VM can be reversed (rescued) by constitutively activating the respective downstream components.
Response 6 : In our previous paper [Han, D. S.; Lee, E. O., Leptin Promotes Vasculogenic Mimicry in Breast Cancer Cells by Regulating Aquaporin-1. Int. J. Mol. Sci. 2024, 25, (10).], we have already confirmed the leptin-STAT3-AQP1-VM axis and described this in the Introduction and Discussion sections.
Comments 7 : The logical sequence of the figures could be improved. The phenotypic experiments validating the "leptin-VM" model (Figs 3, 4, 5) would be more logically placed at the beginning of the results. The mechanistic investigations in Figs 1 and 2, which should build upon the established phenotype, require more robust data for support.
Response 7 : Thank you for your valuable comments. We have already established the leptin-STAT3-AQP1-VM axis in our previous paper [Han, D. S.; Lee, E. O., Leptin Promotes Vasculogenic Mimicry in Breast Cancer Cells by Regulating Aquaporin-1. Int. J. Mol. Sci. 2024, 25, (10).], and this MS confirmed whether Rha inhibits this axis, so it seems better to maintain the current order.
Reviewer 2 Report
Comments and Suggestions for Authors
An original article which is focused on an attempt to substantiate and create a new method for treating patients with triple-negative breast cancer (TNBC) is submitted for review.
The authors' concept is that obesity plays an important role in creating conditions for development of breast cancer by facilitating the process of vasculogenic mimicry.
This is due to an increase of the leptin concentration in obese patients due to its production by adipocytes. Leptin phosphorylates STAT3 causing its translocation to the cell nucleus which leads in its turn to stimulation of the TNBC progression. So, it seems relevant to find a method for disrupting the leptin-STAT3- Aquaporin-1 (AQP1) signal pathway for decreasing the ability of tumor cells to vasculogenic mimicry and, respectively, to adhesion and invasion, and thus diminishing the TNBC malignant potency.
Rhapontigenin, the stilbene derivative, was selected as an agent with potency to disrupt this signal pathway. It possesses some properties which are useful and positive in anti-cancer fight: antioxidative, anti-inflammatory, and antitumor.
The Abstract correctly reflects the content of the article.
The Introduction section, based on the cited literature, describes the problems of TNBC, its relation to obesity, components of the leptin-STAT3-AQP1 signaling pathway and Rhapontigenin, which was used in the study as a therapeutic agent.
Materials and Methods section is written clearly and concisely. But the data obtained from this section is enough for reproducing the experiments and getting similar results.
The Results section includes adequate illustrative material formed in Figures which allow easily understand the performed experiments and the obtained data.
The Discussion section is based on the comparison between the results of the reviewed study and the data from modern literature. The comparative analysis confirmed relevance of the performed experiments and their conclusions.
The manuscript is finalized with short conclusion which generalizes the presented material.
We join the authors in their hope that disrupting leptin-STAT3-AQP1signal pathway by including Rhapontigenin into the treatment scheme for TNBC patients, especially those with obesity, would cause the decrease of disease severity due to diminishing intensity of the vasculogenic mimicry. It is quite possible that this therapeutic agent will promote the sensitivity to anti-angiogenic therapy.
The presented material is of great interest to various specialists: oncologists, physicians, obesity specialists, biochemists, genetics, physicians, and biologists of other profiles.
The manuscript is written in good scientific English and may be recommended for publication in the Biomedicines in the presented form.
Author Response
An original article which is focused on an attempt to substantiate and create a new method for treating patients with triple-negative breast cancer (TNBC) is submitted for review.
The authors' concept is that obesity plays an important role in creating conditions for development of breast cancer by facilitating the process of vasculogenic mimicry.
This is due to an increase of the leptin concentration in obese patients due to its production by adipocytes. Leptin phosphorylates STAT3 causing its translocation to the cell nucleus which leads in its turn to stimulation of the TNBC progression. So, it seems relevant to find a method for disrupting the leptin-STAT3- Aquaporin-1 (AQP1) signal pathway for decreasing the ability of tumor cells to vasculogenic mimicry and, respectively, to adhesion and invasion, and thus diminishing the TNBC malignant potency.
Rhapontigenin, the stilbene derivative, was selected as an agent with potency to disrupt this signal pathway. It possesses some properties which are useful and positive in anti-cancer fight: antioxidative, anti-inflammatory, and antitumor.
The Abstract correctly reflects the content of the article.
The Introduction section, based on the cited literature, describes the problems of TNBC, its relation to obesity, components of the leptin-STAT3-AQP1 signaling pathway and Rhapontigenin, which was used in the study as a therapeutic agent.
Materials and Methods section is written clearly and concisely. But the data obtained from this section is enough for reproducing the experiments and getting similar results.
The Results section includes adequate illustrative material formed in Figures which allow easily understand the performed experiments and the obtained data.
The Discussion section is based on the comparison between the results of the reviewed study and the data from modern literature. The comparative analysis confirmed relevance of the performed experiments and their conclusions.
The manuscript is finalized with short conclusion which generalizes the presented material.
We join the authors in their hope that disrupting leptin-STAT3-AQP1signal pathway by including Rhapontigenin into the treatment scheme for TNBC patients, especially those with obesity, would cause the decrease of disease severity due to diminishing intensity of the vasculogenic mimicry. It is quite possible that this therapeutic agent will promote the sensitivity to anti-angiogenic therapy.
The presented material is of great interest to various specialists: oncologists, physicians, obesity specialists, biochemists, genetics, physicians, and biologists of other profiles.
The manuscript is written in good scientific English and may be recommended for publication in the Biomedicines in the presented form.
Response : Thank you for your valuable comments.
Reviewer 3 Report
Comments and Suggestions for Authors
The manuscript examines rhapontigenin’s inhibitory effect on leptin-induced vasculogenic mimicry in TNBC cells via the STAT3–AQP1 axis. The topic is relevant, and the experimental approach is generally solid, but several issues limit impact. Novelty is overstated; the introduction should clarify what is new compared to prior leptin–VM studies and remove redundancy. Design is restricted to two cell lines without in vivo validation, reducing generalizability. Mechanistic evidence is incomplete—no knockdown or rescue experiments confirm causality. Statistical rigor is weak: reliance on Student’s t-test for multiple comparisons is inappropriate; ANOVA should be applied. Figures lack quantitative clarity and detailed legends.
Line-by-line suggestions:
- Line 88:Add: “Experiments were performed in triplicate; statistical analysis employed ANOVA for multiple group comparisons.”
- Line 120 (Table 1):Include antibody catalog numbers in a separate column for clarity and add a note on validation status.
- Figures 1–4:Improve resolution; add scale bars and replicate numbers in figure legends. Include quantitative bar graphs with error bars for VM and invasion assays.
- Line 339:Add: “Future studies should include in vivo validation and combination therapy approaches.”
- Add new figure:A schematic summarizing the leptin–STAT3–AQP1 axis and Rha’s inhibitory points for visual clarity.
- Add supplementary table:Raw data for RT-qPCR and ChIP assays with fold-change values and p-values.
Minor editing is required
Author Response
The manuscript examines rhapontigenin’s inhibitory effect on leptin-induced vasculogenic mimicry in TNBC cells via the STAT3–AQP1 axis. The topic is relevant, and the experimental approach is generally solid, but several issues limit impact. Novelty is overstated; the introduction should clarify what is new compared to prior leptin–VM studies and remove redundancy. Design is restricted to two cell lines without in vivo validation, reducing generalizability. Mechanistic evidence is incomplete—no knockdown or rescue experiments confirm causality. Statistical rigor is weak: reliance on Student’s t-test for multiple comparisons is inappropriate; ANOVA should be applied. Figures lack quantitative clarity and detailed legends.
Response : In our previous paper [Han, D. S.; Lee, E. O., Leptin Promotes Vasculogenic Mimicry in Breast Cancer Cells by Regulating Aquaporin-1. Int. J. Mol. Sci. 2024, 25, (10).], we have already confirmed the leptin-STAT3-AQP1-VM axis using STAT3 inhibitors and AQP1 overexpression/siRNA, and described this in the Introduction and Discussion sections.
We received English editing before submitting our paper.
Line-by-line suggestions:
Comments 1 : Line 88:Add: “Experiments were performed in triplicate; statistical analysis employed ANOVA for multiple group comparisons.”
Response 1 : According to your comments, we have revised.
Comments 2 : Line 120 (Table 1):Include antibody catalog numbers in a separate column for clarity and add a note on validation status.
Response 2 : The company name, dilution factor, and product number are listed in the Table 1.
Comments 3 : Figures 1–4:Improve resolution; add scale bars and replicate numbers in figure legends. Include quantitative bar graphs with error bars for VM and invasion assays.
Response 3 : According to your comments, we have revised.
Comments 4 : Line 339:Add: “Future studies should include in vivo validation and combination therapy approaches.”
Response 4 : According to your comments, we have revised.
Comments 5 : Add new figure:A schematic summarizing the leptin–STAT3–AQP1 axis and Rha’s inhibitory points for visual clarity.
Response 5 : According to your comments, we have added Figure 6.
Comments 6 : Add supplementary table:Raw data for RT-qPCR and ChIP assays with fold-change values and p-values.
Response 6 : According to your comments, we have added Table A1-4 in appendix section.
Round 2
Reviewer 1 Report
Comments and Suggestions for Authors The authors' response indicates that the mechanism mentioned in this paper—specifically, the leptin-STAT3-AQP1-VM axis—has already been validated in their previous research [Han, D. S.; Lee, E. O., Leptin Promotes Vasculogenic Mimicry in Breast Cancer Cells by Regulating Aquaporin-1. Int. J. Mol. Sci. 2024, 25, (10).], as described in the Introduction and Discussion sections. This confirms that the current study does not explore any novel mechanisms. The only innovation presented here is the finding that Rha inhibits VM, accompanied by reductions in p-STAT3 and AQP1 activity. Therefore, this work constitutes a pharmacological study focusing on Rha in the context of VM. As a pharmacological study, it should delve into the specific molecular targets of the drug rather than merely describing the inhibition of pathway activity. Moreover, the experimental design ought to systematically investigate the effects of concentration gradients and time gradients on both the VM phenotype and the underlying mechanisms. The current study, however, employs only a single concentration and a single time point for Rha treatment, which is insufficient to provide compelling evidence at the pharmacological level. Taking all into consideration, the study largely extends previous conclusions without introducing mechanistic novelty. As a pharmacological investigation, it lacks adherence to fundamental principles of pharmacological experimental design. Thus, I maintain my original critique that "the data presented in the manuscript are relatively limited, and the validation of both the phenotype and the mechanism is insufficient."Author Response
Comments : The authors' response indicates that the mechanism mentioned in this paper—specifically, the leptin-STAT3-AQP1-VM axis—has already been validated in their previous research [Han, D. S.; Lee, E. O., Leptin Promotes Vasculogenic Mimicry in Breast Cancer Cells by Regulating Aquaporin-1. Int. J. Mol. Sci. 2024, 25, (10).], as described in the Introduction and Discussion sections. This confirms that the current study does not explore any novel mechanisms. The only innovation presented here is the finding that Rha inhibits VM, accompanied by reductions in p-STAT3 and AQP1 activity. Therefore, this work constitutes a pharmacological study focusing on Rha in the context of VM. As a pharmacological study, it should delve into the specific molecular targets of the drug rather than merely describing the inhibition of pathway activity. Moreover, the experimental design ought to systematically investigate the effects of concentration gradients and time gradients on both the VM phenotype and the underlying mechanisms. The current study, however, employs only a single concentration and a single time point for Rha treatment, which is insufficient to provide compelling evidence at the pharmacological level. Taking all into consideration, the study largely extends previous conclusions without introducing mechanistic novelty. As a pharmacological investigation, it lacks adherence to fundamental principles of pharmacological experimental design. Thus, I maintain my original critique that "the data presented in the manuscript are relatively limited, and the validation of both the phenotype and the mechanism is insufficient
Response : We acknowledge that our research structure has shortcomings. We simply examined whether there were drugs that inhibited the already proven VM mechanism, and the result was rha. We conducted this study because there had been no previous research on rha in this area. While the concentration and time used in the experiment were single-point measures, we used two cell lines to compensate for this. However, we are fully aware that further research is needed, including animal testing and studies on combination therapy with existing antiangiogenic inhibitors. Simply assessing the efficacy of Rha at the pharmacological level would be difficult. Thank you for your valuable feedback. We will carefully consider this issue in our future research. Thank you.
Round 3
Reviewer 1 Report
Comments and Suggestions for Authors
Thank you for the response. I must point out that the use of two cell lines is a standard and minimal requirement for in vitro studies to ensure findings are not cell-line-specific. It is considered a baseline experimental design, not a compensatory strength for the lack of concentration and time gradients. Therefore, this does not mitigate the fundamental concerns regarding the pharmacological depth of the study. I reiterate that the experimental design remains insufficient to support the pharmacological conclusions the authors wish to draw.
Author Response
Comment : Thank you for the response. I must point out that the use of two cell lines is a standard and minimal requirement for in vitro studies to ensure findings are not cell-line-specific. It is considered a baseline experimental design, not a compensatory strength for the lack of concentration and time gradients. Therefore, this does not mitigate the fundamental concerns regarding the pharmacological depth of the study. I reiterate that the experimental design remains insufficient to support the pharmacological conclusions the authors wish to draw.
Response : Thank you for your time. Based on your advice, we will carefully consider designing the experiment more appropriately to support the pharmacological conclusions about the drug. Thank you.